# Recycling of Aluminum-Based Composites Reinforced with Boron-Tungsten Fibres

**DOI:** 10.3390/ma15093207

**Published:** 2022-04-29

**Authors:** Viktors Mironovs, Yulia Usherenko, Irina Boiko, Jekaterina Kuzmina

**Affiliations:** 1Scientific Laboratory of Powder Materials, Faculty of Mechanical Engineering, Transport and Aeronautics, Riga Technical University, 6A Kipsalas Str., Lab. 319, LV-1048 Riga, Latvia; viktors.mironovs@rtu.lv (V.M.); yuliausherenko@gmail.com (Y.U.); 2Institute of Mechanics and Mechanical Engineering, Faculty of Mechanical Engineering, Transport and Aeronautics, Riga Technical University, 6B Kipsalas Str., Room 406, LV-1048 Riga, Latvia; irina.boiko@rtu.lv

**Keywords:** composite materials, powder metallurgy

## Abstract

High strength fibres of carbon, boron, silicon carbide, tungsten, and other materials are widely used to reinforce metal matrix composite materials. Carbon and boron fibers are usually used to reinforce light alloys based on aluminum and magnesium. Products made from these materials are characterized by high strength and rigidity and can be used for a long time. Technological waste containing such fibres are hazardous to the environment because they are durable and have needle-like and other sharp shapes. Therefore, they must be disposed of with extreme care. A significant incentive for the processing and reuse of waste composites of this type is the relatively high cost of production of the primary fibre and the material as a whole. With the increase in the production of such materials in recent years, the need to recycle composite waste is becoming increasingly important. Three main options for primary processing are used to prepare composites for their subsequent use. They are mechanical, thermal, and chemical grinding technologies. One of the actual and practical areas of processing technology is the method of powder metallurgy. This paper presents the main stages of processing composite materials based on an aluminium matrix and B-W fibres to obtain powder compositions. The results of the studies showing the possibility of the effective use of the obtained crushed waste to manufacture concrete products and the production of cutting and grinding tools are presented.

## 1. Introduction

One trend indicates the general progress and preference for environmentally friendly, energy-efficient, and sustainable technologies in producing materials and structures. This trend requires a shift from traditional materials, such as aluminium and steel, towards more efficient (by specific module GPa·kg^−1^·m^−3^ and specific stiffness MPa·kg^−1^·m^−3^ [1]) alternatives: lightweight composites based on ceramics [2], polymer matrix [3,4,5], metal matrix [6,7], and hybrid materials [8,9]. Most composite materials are broadly fibre-reinforced composites [10,11]. These composites have been long and widely used in the construction, automotive, aviation, and marine industries [12].

The missing link necessary to complete the cycle of applicability of these composites is the requirement for the end-of-life recycling [13]. The primary purpose is to highlight the available technologies for recycling fibre-reinforced composites and the socio-economic and other environmental implications, using aluminium-based composites as an example. The cited articles cover the period from the 1990s to the present, embracing the period when the impact of composites on the environment has become globally significant [14,15,16].

It is known that composite materials on a metal base consist of the metal matrix (Al, Mg, Ni, and their alloys) reinforced by high-strength fibres (fibre materials) [17] or fine-dispersed refractory particles [18], not dissolved in the base metal (dispersion reinforced materials) [19].

Fibrous composite materials with metal matrices differ from conventional alloys by their high values of time resistance, endurance limit and elastic modulus, and a reduced tendency to cracking [20]. In addition, using fibrous composite materials with metal matrices materials increases the structure’s stiffness while reducing its metal capacity [21,22]. 

Due to their high strength (σ_b_, σ^−1^), low weight, heat resistance, and other valuable properties, FCMMs are effectively used in modern industry, especially in aircraft engineering [23], shipbuilding [24], and instrumentation [25]. An increase in the heat resistance of CM is achieved by reinforcing them with tungsten or molybdenum wires [26]. Metal fibres are also used in those cases where high thermal and electrical conductivity are required. Boron and carbon fibres and refractory compound fibres with high strength and modulus of elasticity, such as carbides, nitrides, borides, carbonitrides, and oxides, are used for strengthening metals. Boron, carbon, or silicon carbide fibres are often used to enhance high-strength steels and amorphous alloys [27,28]. At the same time, boron fibres have the highest strength values (σ_b_ = 2500–3500 MPa) and modulus of elasticity (E = 380–420 GPa) [29]. The properties of the obtained fibrous composite materials with metal matrices depend on the technological parameters of the manufacturing processes, properties, and volume content of the fibres [30,31].

Al-B composites (Table 1) have high tensile strength values (σ_b_), endurance limit of materials (σ^−1^), high specific strength (σ_b_/γ) and modulus of elasticity (E/γ).

Boron aluminium (Al-B) is 3.5 times lighter than aluminium and twice as potent, saving significant weight [32,33]. Al-B has two times higher values of specific strength and stiffness, compared, for example, with titanium, which makes it possible to use it to manufacture elements of aircraft structures [34]. Boron fibres are produced by chemical deposition from the vapour phase of boron on a tungsten carbon wire (core) with a diameter of 10–12 microns [35]. Consequently, boron fibres have a large diameter—100–200 microns. It should be noted that these materials are brittle and sensitive to surface damage. B-W fibres are used as fillers in fibrous composite materials with metal matrices with a metal matrix [36]. Boron is an essential element in the production of high-strength materials. However, the various applications of boron and its compounds may cause increasing hazards [37] for environmental contamination and cause health problems [38].

The apparent need for the efficient use of raw materials, supported at the European Union level, emphasises the recycling of products and materials. Therefore, the modern research and engineering community is paying particular attention to the efficient recycling of end-of-life energy storage devices [39,40], metals that are critical raw materials [41,42], used adsorbents [43,44], and other valuable materials [45,46] for which recycling into the production and consumption cycle is economically viable.

Considering the importance of tungsten-containing composite materials, this work aims to determine methods for the efficient processing of Al-W-B fibrous composite materials with metal matrices.

## 2. Processing of Al-W-B Technological Waste into Powder and Their Reuse

### 2.1. Recycling of Composite Materials

The increase in the production and consumption of metal matrix composites invariably entails an increase in waste generated both at the production stage and at the end of the product’s service life. In this regard, the utilisation and/or processing of metal matrix composites becomes more and more acute every day [47]. Several countries and international organisations insist on considering the development of non-waste technologies, e.g., closed-cycle production, as a strategic direction [4]. This does not mean that there would be no waste at all. Still, its amount should be minimised and provided for in by-products or the possibility of reuse with the parallel development of the most rational recycling programs. According to European legislation (Directive 2008/98/EC), there is a priority of waste management methods, in which the lowest priority method is incineration and others are stored [48]. The most acceptable option for the disposal of metal matrix composites is their processing and the reuse of the resulting products. This fully applies to fibrous composite materials with metal matrices containing boron-tungsten fibres. The low susceptibility of such composites to deterioration under environmental factors is an indisputable advantage when considering aspects of their operation and maintenance. Fibrous composite material with metal matrices waste stored at landfills decomposes extremely slowly and becomes a factor in the environmental pollution [49]. Here we can distinguish three main groups of disposal methods (Figure 1): based mechanical processing; involving the chemical removal of matrix material with the partial/complete preservation of fibres; based on a thermal (full or partial) decomposition of the matrix and fibres.

The physical methods of processing fibrous composite materials with metal matrices [50] are characterised by minimal environmental impact. First, this group includes the mechanical process by grinding fibrous composite materials with metal matrices. In this case, large fragments or fractions obtained during grinding can be of different sizes (fineness) depending on the method of further application.

Waste in the form of plates and pipes processed by this method can be used extremely successfully, to create new products and materials. However, one should consider the energy consumption of the mechanical method of grinding fibrous composite materials with metal matrices reinforced with high-strength fibres [51].

Chemical recycling methods for fibrous composite materials with metal matrices [52] are characterised by high energy efficiency. Chemical action is selected directly for a specific combination of fibre and matrix. This allows you to get the maximum possible amount of product suitable for reuse at the output stage with a minimum investment of time and resources.

Studies [53] show that etching the surface of the metal and other fibres is promising. It can effectively remove the matrix material and return the fibre to the production of composites.

Of the thermal methods, the method of melting the matrix is of the greatest interest [54]. This is especially interesting for fibrous composite materials with metal matrices with an aluminum matrix. It is also possible to use pyrolysis undertaken in an oxygen-free environment. Today, this method is common for recycling reinforced composites [54,55]. The peculiarities of thermal processes include the requirements for heat resistant fibres: melting and pyrolysis are carried out at elevated temperatures (from 300 to 1000 °C, depending on the variety). Therefore, it applies only to heat-resistant basalt, tungsten and carbon fibres. In this case, some loss of fibre strength is possible. The reinforcing materials obtained at the output are most often suitable for reuse in reinforcing lightly loaded thermoplastics and concretes [56,57].

When considering the problem of recycling fibrous composite materials with metal matrices, it is necessary to develop methods for processing composites and create the possibility of recycling these products. The existing developments in this area open up vast opportunities for creating products with the desired mechanical characteristics with the processing problem pre-solved.

Some research [49,50] showed that recycling out-of-use fibrous composite materials with metal matrices is a rather complicated technological process. These materials need to be pre-sorted for reuse into homogeneous categories with high purity. The proper recycling technology must be selected as well. It is essential to achieve maximum recycling of all fibrous composite materials with metal matrices components [58]. Even low-grade fibrous composite materials with metal matrices fractions contain the potential to create new valuable products [35,59].

### 2.2. Technological Waste of Al-W-B

The main types of technological waste used are Al-W-B products, pipe ends after extrusion processing, sheets and plates, and W-B fibre prepregs in an aluminium matrix (Figure 2). W-B fibres are made by decomposition of boron chloride or boron bromide on a hot W filament. This technology is quite widely known by [60]. The microhardness of the B-W fibres is very high (from 56 to 70 GPa) compared to the matrix material (2.2–4.5 GPa) [29].

Prepregs are produced by spraying aluminium onto preformed W-B fibres. For the manufacture of Al-W-B composites, technology is often used to impregnate W-B fibers with aluminium [35]. In this case, the interaction of molten aluminium with boron fibre should be minimal. The technology involves the hot rolling of layer-by-layer prepregs and is widely used. In this case, the hot pressing method produces a composite with W-B fibres.

The macro and microstructure of Al-W-B are presented on Figure 3.

### 2.3. Processing Al-W-B Waste into Powder Material 

Crushing and disintegration methods are used to obtain fibrous powder Al-W-B [32]. Crushing is carried out in several stages to obtain a product with a given particle size. Using scanning electron microscopy (SEM) combined with energy dispersive X-ray spectroscopy in [33] to determine the composition of Al-W-B, it was found that the matrix material is an aluminium alloy, which consists of Al (90.34%), Mg (5.81%), Mn (2.42%), Fe (1.32%), and the powder material obtained has the composition: B, 47.05%; Al, 43.26%; W, 5.06%; Mg, 2.78%; Mn, 1.16%; Fe, 0.63%. In this powder, B and W had a purity of 99.9. The particle sizes of Al-W-B ranged from approximately 2 μm to 100 μm (Figure 4). The microhardness of the particles varies from 2–3 GPa (Al); 25–30 GPa (B), to 60–70 GPa (W).

During crushing, one can observe the separation of fibre from the Al matrix, the uneven distribution of short and long particles of boron fibre in the mixture, and at the same time, pronounced elements of the crystal shape of boron fibre. In addition, there is axial destruction of the W-B fibre and stratification (Figure 5).

The grinding in the disintegrator makes it possible to obtain a fraction of the powder component in a given range (Figure 6). 

Further grinding of the material into powder was carried out in a disintegrator [36], which makes it possible to obtain finer fractions of the material (Figure 7).

## 3. Preparation of Secondary Products from Al-W-B Waste 

The unique properties of the Al-W-B material after milling, especially the high hardness of the fibers, allow us to make suggestions for some technological possibilities for its application.

### 3.1. Making Grinding Tools

One of the technologies of using composite powders Al-W-B and W-B is the production technology of grinding and polishing tools. Two variants have been successfully tested to produce elements with high abrasiveness in work [35]. The first variant involves the use of direct cutting from the waste grinding elements (Figure 8).

To determine the coefficient of friction under sliding friction conditions, the test was performed on a Anton Paar GmbH (Graz, Austria) pin-on-disk tribometer TRB-S-EE-000. The measurement settings are shown in Figure 9.

The measurement of the coefficient of friction of a sample that has been mechanically ground with P400 sandpaper and degreased with 96% ethanol before measurement. 

Following the recommendations [61], a vertical friction machine was tested on abrasive grinding elements when grinding granite and concrete. Grinding parts were installed in unique heads and fixed with epoxy resin. Main parameters: number of head revolutions was 400–600 rpm; grinding speed (circumferential)—8–12 m·s^−1^; the pressure of the tool to the machined surface—0.04 MPa. (Figure 10).

A preliminary study of the material properties, having boron-containing fibres, allowed us to conclude the possibility of applying these materials in the manufacturing of grinding elements for processing various materials. However, finer and more isotropic fibres should be used, and the density and strength of the details should be increased. 

### 3.2. Possibilities of Using Al-W-B for High-Strength Concrete

Aluminium matrix composites are lightweight materials with increased ductility and wear resistance. Their strength can be significantly improved by adding suitable reinforcing phases (particles, short or continuous fibres. One such composite material could be high-strength concrete with Al-W-B fibres. In the work [38], the results are given of a study of the effect of different concentrations of Al-W-B powder on properties. We used alite containing white Portland cement, granite stone, sand, microsilica and superplasticiser, and crushed fibrous Al-W-B powder in constructing the concrete mix (Figure 11). It was noticed that even a significantly increased concentration of Al-W-B (up to 200 kg·m^−3^ of concrete volume) did not drastically change the required water-cement ratio. Testing the properties of the material showed an increase in the strength properties of the material. Tests showed that the density of the samples did not exceed 1684 kg/m3. At the same time, the strength indicators were relatively high and were: for bending 5.17–6.31, and for compression 17.4–23.6 MPa

However, researchers noted a deterioration in the surface of the product and a decrease in the flowability of the dry mixture. Using chopped WB fibres without an aluminium phase can provide better results.

### 3.3. Possibilities of Using Al-W-B for Nuclear Energy

Boron is often included in the composition of structural materials from which neutron shielding elements are made. Currently, parts for such containers are made of boron steel containing no more than 6–8 at.% boron. Existing estimates [62] show that when replacing boron steel with boron-containing materials based on aluminium reinforced with boron fibres, their weight is reduced by 10–30%, and the neutron-absorbing properties are increased. We prepared aluminum-based metal matrix composites containing up to 30 wt.% filler from boron-tungsten powder by mechanical activation in a high-energy dispersant. Bulk samples were prepared from the obtained powder using static and dynamic compaction methods for subsequent studies. Structural and microscopic methods were used to study the resulting powder samples; the dependences of phase and structural parameters (phase composition, filler particle size, matrix, and filler grain sizes) on the initial design and processing parameters were revealed. At the same time, it is essential to study the uniformity of the distribution of solid particles in a soft aluminium matrix.

### 3.4. Obtaining Dispersion-Hardened Materials from Powders with B-W Particles

It is also possible to remove the aluminium matrix. A chemical method was used in the work [33] (Figure 12) for removal. 

The second option considers the technology of powder metallurgy. A mixture containing crushed fibrous powders W-B and a powder base based on copper, iron, or other metals were prepared, compacted by pressing and subjected to sintering (Figure 13). For pressing, the magnetic pulse pressing method was used [60]. The microstructure of copper-based samples is shown in Figure 14.

## 4. Discussion and Conclusions

Process waste materials containing metal fibres, particularly Al-W-B, are hazardous to the environment and must be disposed of with extreme care. However, the high cost of producing primary fibres and material requires compulsory processing and recycling.

One of the directions for the secondary use of boron-containing wastes is powders and fibres, which can create new composite materials, as shown in many works.

Analysing numerous sources of information and the authors’ own research, we can definitely say that the problem of recycling boron-tungsten composites is very relevant. At the same time, several main promising areas for their secondary use can be distinguished: -Obtaining fragments from the original Al-W-B material by cutting large elements into smaller ones and using the latter to perform grinding-rolling-polishing work;-Obtaining a new type of fibrous material Al-W-B by sequential grinding of waste and using it to create new composite materials, in particular, concrete;-Obtaining W-B fibers from the original Al-W-B after removing the aluminum matrix and subsequent grinding of the product;-Production of Al-W-B and W-B powders and their use in the charge for the manufacture of a wide range of products using powder metallurgy methods.

The research has shown that W-B-containing materials require additional energy costs to effectively reuse their waste products. This is due to the fact that for the processing of Al-W-B waste materials with an initial high hardness of the components, it is necessary to use modern equipment, such as diamond cutting, electric spark processing, modern mills, disintegrators, and dispersers. For the implementation of the grinding processes of W-B-containing materials, equipment with increased wear resistance of the grinding elements and the working chamber is required. The process of separating the matrix material also requires the use of special chemical methods. The manufacture of products from composite powders containing sharp and hard fine W-B particles requires a special approach when pressing since technological equipment is subject to increased wear.

Particular attention in all technological processes and research operations should be paid to the safety of personnel, due to the unacceptability of receiving needle-shaped highly solid, and small products from W-B-containing materials into the respiratory tract and onto the skin.

## Figures and Tables

**Figure 1 materials-15-03207-f001:**
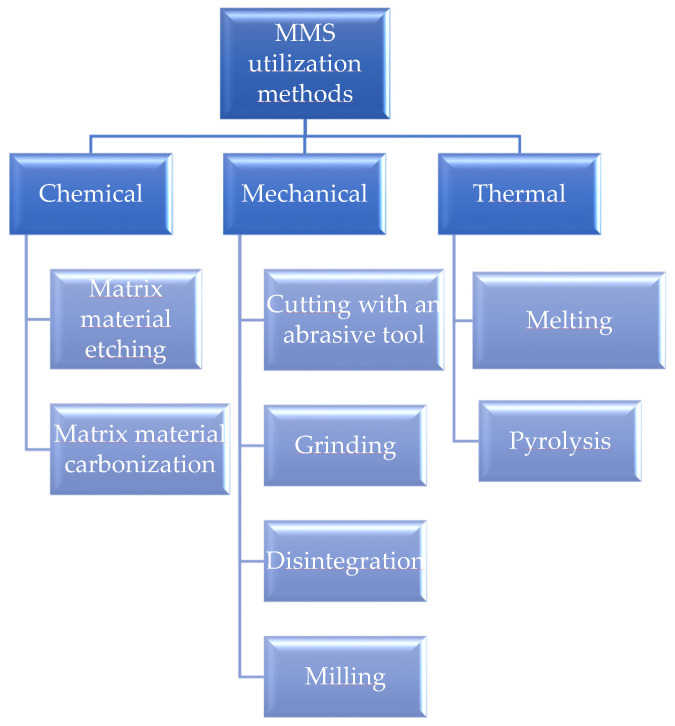
Fibrous composite materials with metal matrices disposal methods.

**Figure 2 materials-15-03207-f002:**
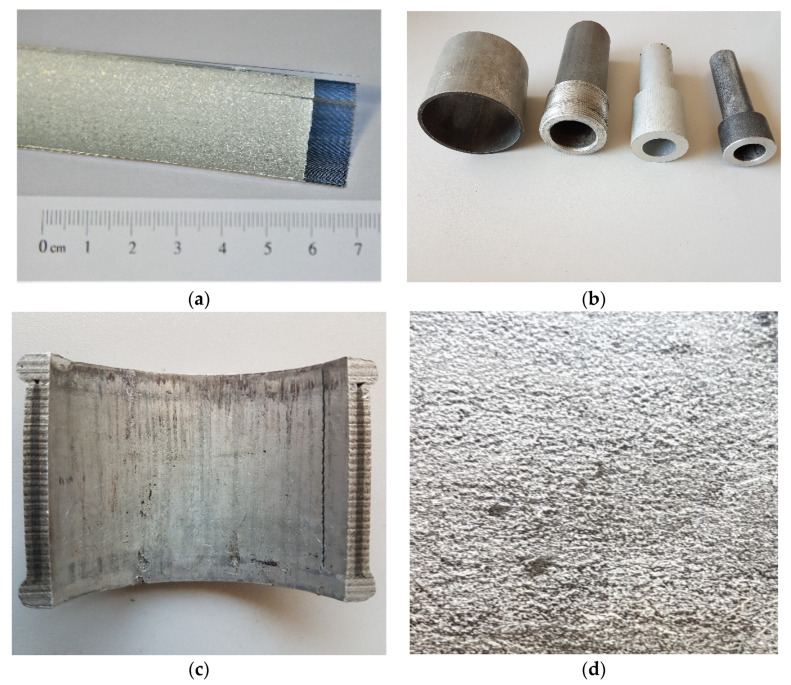
Fragments of (B-W/fibrous composite materials with metal matrices): prepregs from W-B fibres with subsequent aluminium sputtering (**a**); tubes after hot pressing of prepregs (diameter 20–100 mm (**b**); also—in section (**c**); multilayer plate of B-W fibres in a matrix of aluminium alloy (**c**) thickness—4 mm (**d**).

**Figure 3 materials-15-03207-f003:**
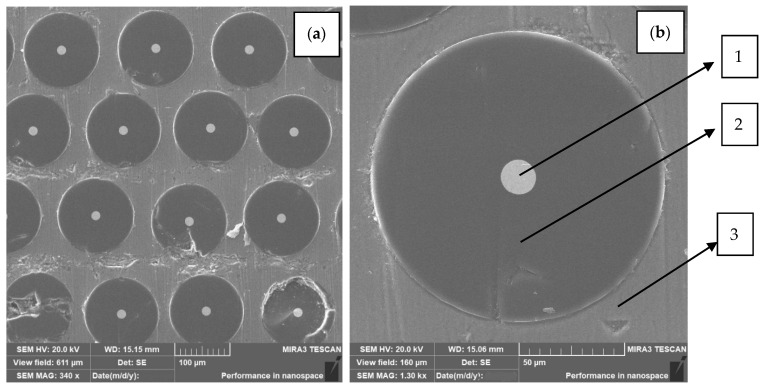
Al-W-B microstructure (**a**,**b**): 1—tungsten wire rod 10–12 µm, 2—boron coating with a diameter of 70–100 µm; 3—matrix material aluminium.

**Figure 4 materials-15-03207-f004:**
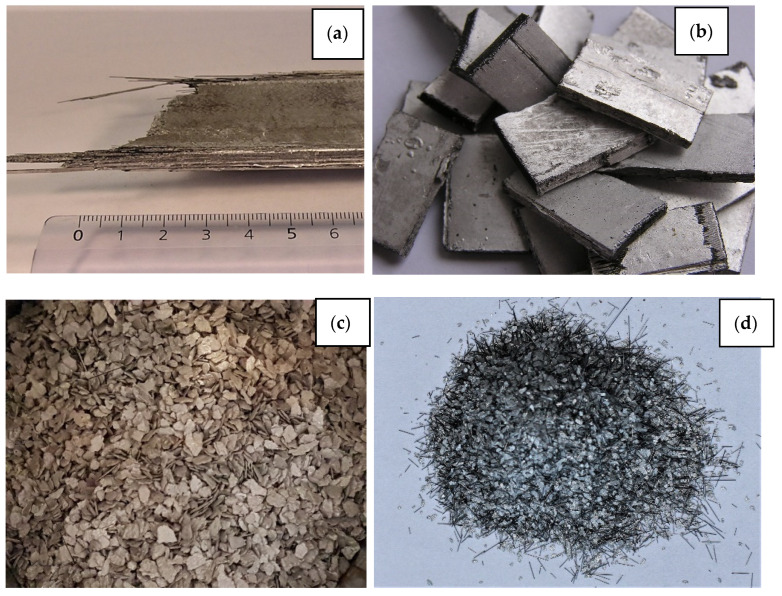
Stages of Al-W-B waste shredding: breaking (**a**) and cutting the plates (**b**); shredding and milling the material (**c**,**d**).

**Figure 5 materials-15-03207-f005:**
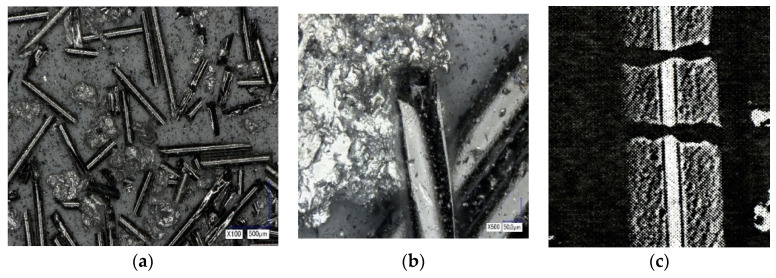
Microstructure of Al-W-B composite after grinding in a vibrating mill (**a**–**c**).

**Figure 6 materials-15-03207-f006:**
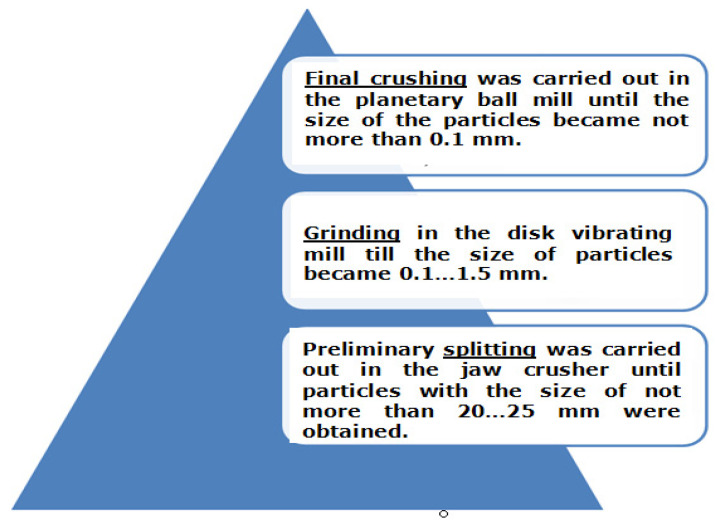
Grinding diagram Al-W-B waste into powder material at various stages.

**Figure 7 materials-15-03207-f007:**
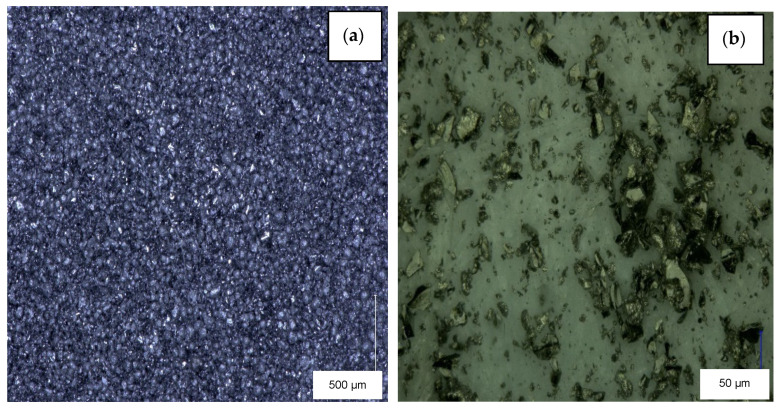
Microstructure of Al-W-B composite powders after grinding in disintegrator (**a**,**b**).

**Figure 8 materials-15-03207-f008:**
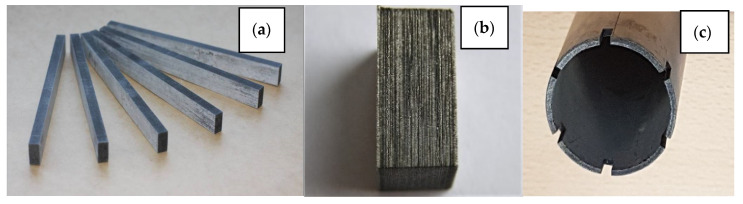
Elements for grinding and polishing devices: honing stones 6 × 10 × 120 mm (**a**,**b**); ring drill Ø 45 mm (**c**).

**Figure 9 materials-15-03207-f009:**
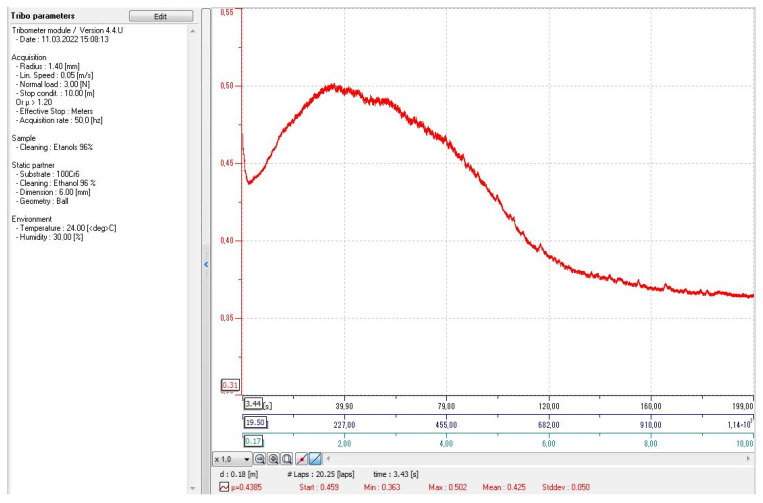
Coefficient of friction versus time, cycles, distance.

**Figure 10 materials-15-03207-f010:**
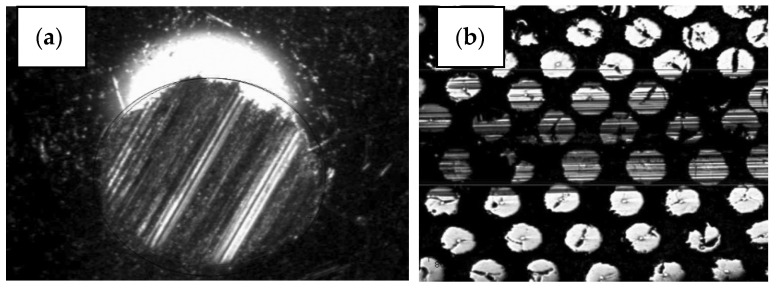
Wear images of the counterparty and the body under study (**a**) spherical counterpart made of steel 100 Cr 6; (**b**) the body being examined.

**Figure 11 materials-15-03207-f011:**
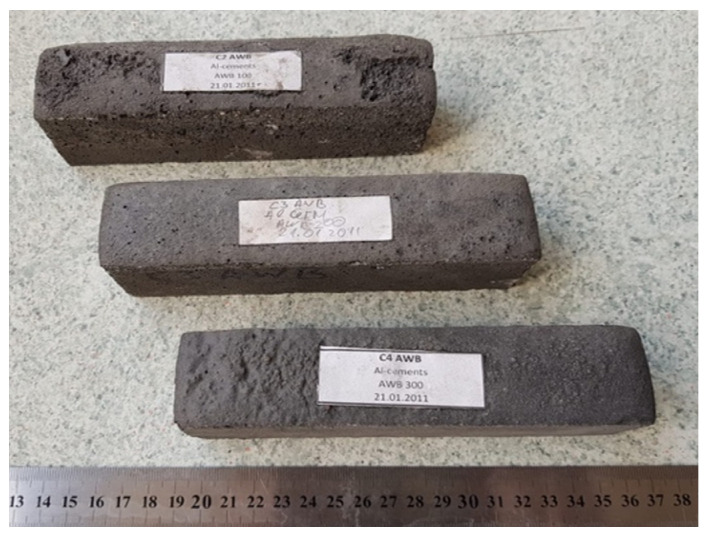
Sample of concrete containing boron 40 × 40 × 160 mm^3^.

**Figure 12 materials-15-03207-f012:**
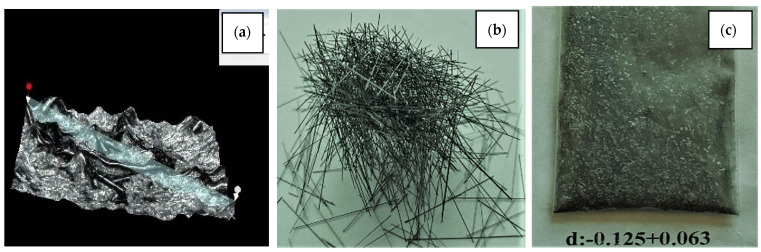
Boron-containing material: in an aluminium matrix (**a**); (W-B) fibres after chemical cleaning from aluminium matrix material: general view (**b**); W-B fibre powder (**c**).

**Figure 13 materials-15-03207-f013:**
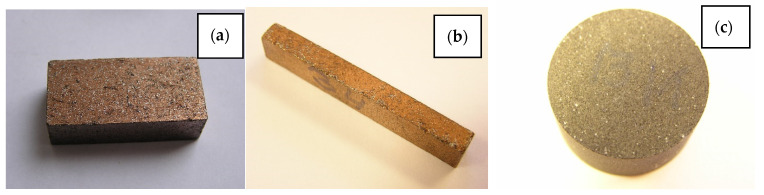
Elements filled with W-B (30–50%) based on copper (**a**,**b**) and iron (**c**) powder after pressing and sintering.

**Figure 14 materials-15-03207-f014:**
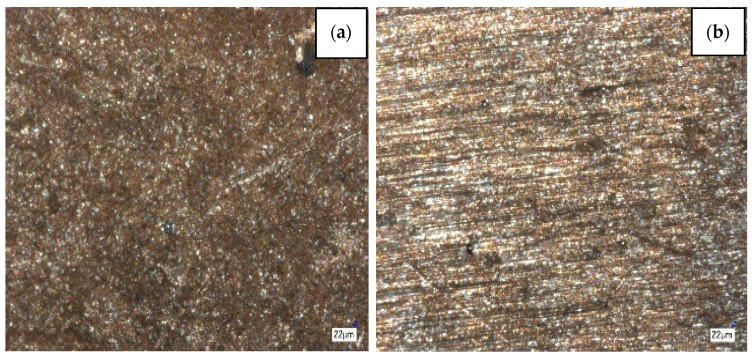
Microstructure of elements based on copper with the inclusion of W-B fibers in a copper matrix (**a**,**b**).

**Table 1 materials-15-03207-t001:** Mechanical properties of metal-based composites [28].

Material	σ_b_	σ^−1^	σ_b_/γ	σ_b_/γ	E/γ
	MPa	MPa	GPA		
Boron aluminium (Al-W-B)	1300	600	220	500	84.6
Boron magnesium (Mg-B)	1300	500	220	590	100.0
Aluminium-carbon (Al-C)	900	300	220	450	100.0
Aluminium-steel (Al-St)	1700	350	110	370	24.4
Nickel-tungsten (Ni-W)	700	150	215	430	60.0

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
