# Peer review of "Recycling of Aluminum-Based Composites Reinforced with Boron-Tungsten Fibres"

_materials, 2022, doi:10.3390/ma15093207_

Round 1
Reviewer 1 Report
Title: Recycling of Aluminum-Based Composites Reinforced with B-W Fibers. After careful evaluation of the submitted manuscript the Reviewer suggests the following points for the improvement of this manuscript.
- Avoid abbreviations or symbols or single letter words in the title of the amnuscript.
- The authors have cited more references in the single line of the manuscript. For example Ref. No [5-8]. This is not the correct way to cite the references. Analyze each and every cited reference thoroughly and give the authors own inference.
- Avoid tables in the Introduction part of the manuscript.
- If the figures are taken from the literature, that to be cited properly.
- Almost all the figures are taken from the published papers in this review. Some figures need to be self-drawn.
- Conclusion part is very poorly written. Summarize the major findings of this review in the conclusion.
Author Response
Thank you very much for your review. We highly appreciate it.
We have made adjustments based on your comments. Please see the attachment.

Reviewer 2 Report
Dear editor in chief This paper have very good information and results.but the authors proposed to do this changes: 1. their are some english mistakes and some sentences with other languages missed in the paper. 2. they should obay standards of paper writing specially in the discussion and result chapter. 3. please bring the test methods of composition testing. 4. please complete the section 3 or Discution chapter. 5. please bring the resultes with numbers from 1 to... . Best wishesAuthor Response
Thank you very much for your review. We highly appreciate it.
We have made adjustments based on your comments. Please see the attachment.

Reviewer 3 Report
The article is only of a review nature - insignificant share of research results carried out within the article, no new knowledge - apart from that contained in the cited publications.
It is recommended to extend the content with:
-a description of recycling methods,
- new/own research results of recycled materials
- examples of applications in specific technical solutions - along with a comparison/indication of their better properties in relation to the current solutions.
Summary and conclusions poorly related to the content of the article.
The article is poorly prepared in terms of editing - both the text and the drawings need to be improved.
High degree of self-citations. Several literature items were not cited in the text of the article.
A number of comments are included in the appendix.

Author Response

(The authors gave the same response as above.)

Reviewer 4 Report
General review: This author explains that high strength fibers of carbon, boron, silicon carbide, tungsten and other materials widely used to reinforce metal matrix composite materials. Carbon and boron fibers are usually used to reinforce light alloys based on aluminum and magnesium. Products made from these materials are characterized by high strength and rigidity and can be used for a long time. Technological wastes containing such fibers are hazardous to the environment because they are durable, have a needle shape and other sharp shapes. Therefore, they must be disposed of with extreme care. A significant incentive for the processing and reuse of waste composites of this type is the relatively high cost of production of primary fiber and the material as a whole. With the increase in the production of such materials in recent years, the need to recycle composite waste is becoming increasingly important. Three main options for primary processing are used to prepare composites for their subsequent use. They are mechanical, thermal, and chemical grinding technologies. One of the essential and practical areas of processing technology is powder metallurgy methods. This paper presents the main directions of processing composite materials based on an aluminum matrix and boron-containing fibers. Many examples of effective recycling of Al-W-B type technological waste materials by grinding them into powder, subsequent pressing, and sintering are given.
Minor revision:
- The authors should mention the fabrication method of the W-B reinforcement in detail, can you mention the ASTM regulation number or any basis knowledges? Or how the W-B-reinforced metal matrix composite was fabricated by casting or any other methods in detail?
- Do the metal matrix composites have any problems? The first sentence in the Introduction is okay: it is very general and induces the interests. Then, it would be better to mention any problems for the next sentence. Thereafter, it is better to mention that you solve these problems using your MMC.
- For the comment of hybrid material in Introduction, I strongly recommend to read and cite the manuscript entitled “Selective compositional range exclusion via directed energy deposition to produce a defect-free Inconel 718/SS 316L functionally graded material”, and to change from the very old references to the new ones as possible.
Author Response

(The authors gave the same response as above.)

Round 2
Reviewer 1 Report
The revised manuscript is improved and the authors are responded well for the queries raised by me in my previous review. Now I recommend this manuscript for publication.
Reviewer 2 Report
Dear Dr.
The revised manuscript now can be accepted.
best wishes
Reviewer 3 Report
The authors took into account most of the comments from the review - though not all (which is a pity).